# Electrochemical Deposition of Fe–Co–Ni Samples with Different Co Contents and Characterization of Their Microstructural and Magnetic Properties

Van Cao Long [1], Umut Saraç [2,*], Mevlana Celalettin Baykul [3], Luong Duong Trong [4], Ştefan Ţălu [5,*] and Dung Nguyen Trong [1,6,*]

1 Institute of Physics, University of Zielona Góra, Prof. Szafrana 4a, 65-516 Zielona Góra, Poland; vancaolong2020@gmail.com
2 Department of Science Education, Bartın University, Bartın 74100, Turkey
3 Department of Metallurgical and Materials Engineering, Eskişehir Osmangazi University, Eskişehir 26480, Turkey; cbaykul@ogu.edu.tr
4 Department of Electronic Technology and Biomedical Engineering, Hanoi University of Science and Technology, Hanoi 100000, Vietnam; luong.duongtrong@hust.edu.vn
5 The Directorate of Research, Development and Innovation Management (DMCDI), Technical University of Cluj-Napoca, 15 Constantin Daicoviciu St., 400020 Cluj-Napoca, Romania
6 Faculty of Physics, Hanoi National University of Education, 136 Xuan Thuy, Cau Giay, Hanoi 100000, Vietnam
* Correspondence: usarac@bartin.edu.tr (U.S.); stefan.talu@auto.utcluj.ro (T.Ş.); dungntsphn@hnue.edu.vn (D.N.T.)

**Abstract:** In this study, to explore the effect of Co contents on the electroplated Fe–Co–Ni samples, three different Fe–Co$_{33}$–Ni$_{62}$, Fe–Co$_{43}$–Ni$_{53}$, and Fe–Co$_{61}$–Ni$_{36}$ samples were electrochemically grown from Plating Solutions (PSs) containing different amounts of Co ions on indium tin oxide substrates. Compositional analysis showed that an increase in the Co ion concentration in the PS gives rise to an increment in the weight fraction of Co in the sample. In all samples, the co–deposition characteristic was described as anomalous. The samples exhibited a predominant reflection from the (111) plane of the face–centered cubic structure. However, the Fe–Co$_{61}$–Ni$_{36}$ sample also had a weak reflection from the (100) plane of the hexagonal close–packed structure of Co. An enhancement in the Co contents caused a strong decrement in the crystallinity, resulting in a decrease in the size of the crystallites. The Fe–Co$_{33}$–Ni$_{62}$ sample exhibited a more compact surface structure comprising only cauliflower–like agglomerates, while the Fe–Co$_{43}$–Ni$_{53}$ and Fe–Co$_{61}$–Ni$_{36}$ samples had a surface structure consisting of both pyramidal particles and cauliflower–like agglomerates. The results also revealed that different Co contents play an important role in the surface roughness parameters. From the magnetic analysis of the samples, it was understood that the Fe–Co$_{61}$–Ni$_{36}$ sample has a higher coercive field and magnetic squareness ratio than the Fe–Co$_{43}$–Ni$_{53}$ and Fe–Co$_{33}$–Ni$_{62}$ samples. The differences observed in the magnetic characteristics of the samples were attributed to the changes revealed in their phase structure and surface roughness parameters. The obtained results are the basis for the fabrication of future magnetic devices.

**Keywords:** cauliflower–like agglomerates; Co contents; crystallinity; Fe–Co–Ni thin film samples; magnetic properties; phase structure; pyramidal particles; roughness parameters

## 1. Introduction

Nanostructured ferromagnetic materials in the form of thin films are widely used in many technological applications and attract great attention because of their good physical and magnetic features [1–5]. To date, many physical and chemical growth techniques have been developed that are utilized in the production process of magnetic thin film samples. Among the growth techniques developed, the electrochemical deposition technique has

been successfully used in computer read/write heads and Micro–ElectroMechanical Systems (MEMS) applications due to its unique features [1,3,6–12]. It is well known that ternary ferromagnetic alloy films are interesting soft magnetic materials due to their high saturation magnetization and low coercive field [4,6]. The conducted studies showed that the Fe, Ni, and Co components in binary Ni–Co, Ni–Fe and Fe–Cu and ternary Ni–Co–Cu, Ni–Fe–Cu, Co–Fe–Cu and Fe–Co–Ni magnetic materials grew by the electrochemical deposition technique on Indium Tin Oxide (ITO) covered glass substrates which can be tuned by controlling the Fe, Ni and Co ion concentrations in the Plating Solutions (PSs), respectively [13–21]. However, the relative compositions of Co and Fe components in the samples were found be higher than those in the PSs for different electroplating parameters [6,13,14,16–24], which is in good agreement with the definition defined by Brenner [25]. In recent years, scientists have flexibly used a variety of methods to study the structural, mechanical, and magnetic properties of nanomaterials and thin films. With the simulation method, the influence of size, heating rate, temperature and annealing time has been successfully studied on the structure, electronic structure, phase transition and mechanical properties of metal Ni [26–28], Fe [29], Al [30,31], Alloy AuCu [32,33], CuNi [34–36], NiAu [37], FeC [38], FeNi [39,40], AgAu [41], AlNi [42], and NiCu [43].

In addition, with the magnetism of nanomaterials and thin films, the influence of nanoparticle size and shell thickness has been successfully studied on the Curie Tc phase transition temperature of Fe nanoparticles [44], the influence of external magnetic field and size on the temperature Neel TN phase transition of $Fe_2O_3$ thin films [45]. The obtained results show that the Neel TN transition temperature is always smaller than the Curie Tc phase transition temperature, the cause of this phenomenon is due to the Topo effect. With the experimental method, the authors have successfully studied the effects of Fe ion concentration in the PS. Furthermore, the deposition potential applied during electroplating process on the chemical composition, some physical properties, and magnetic characteristics of ternary ferromagnetic thin film samples were investigated [21,24]. These experimental studies clearly demonstrated that the surface performance, magnetic and structural characteristics were affected significantly by the deposit composition caused by the variation of the electroplating parameters [21,24]. On the other hand, in a former study, Fe–Co–Ni deposits were electrochemically manufactured on titanium sheets from a chloride–sulfate–tartaric acid medium at different $Co^{2+}/Ni^{2+}$ ion ratios [46]. In another study, Fe–Co–Ni films were electrochemically fabricated on copper substrates from an ammonium–chloride–based PS at different $Co^{2+}/Ni^{2+}$ ion ratios [47]. In addition, in a very recent study, nanocrystalline Ni–Co–Fe coatings were electroplated on copper plates from a sulfate–citrate PS at different Co ion concentrations [3]. To make magnetic Fe–Co–Ni thin films, researchers can use many methods such as evaporation and electrochemical deposition [48]. Among these, thin films obtained by the electrochemical deposition method have very high uniformity. However, in this work, Fe–Co–Ni samples were electroplated on ITO substrates and the Co contents in the samples were tuned by the amount of Co ion concentration in the sulfate-based PS. The structure, morphology, and magnetic characteristics of the resultant samples were discussed with respect to their Co contents. The results showed that the crystallinity, crystallite size, phase structure, particle shape, particle size, magnetic properties, and roughness parameters of the samples are strongly dependent on their Co contents. The obtained results will be applied in different fields of science and technology.

## 2. Materials and Method

The Fe–Co–Ni thin film samples were produced from PSs composed of Ni sulfate (0.07 M), Fe sulfate (0.0020 M), boric acid (0.1 M), and various Co sulfate concentrations (0.016 M, 0.024 M and 0.040 M). The samples were deposited galvanostatically at the same current density of –10 mA/cm$^2$ from freshly prepared PSs (pH value was 5.2 ± 0.1 and temperature was 22 ± 1 °C) without stirring. The electroplating processes were performed by employing a three-electrode system. A platinum sheet was utilized as a counter electrode,

whereas a Saturated Calomel Electrode (SCE) was served as a reference electrode. The samples were grown on ITO coated glass substrates used as a working electrode. Before the plating process, the substrates were first rinsed in an acetone solution and then in ethanol solution. After that, the substrates were cleaned by an ultrasonic bath using deionized water. The crystal structure was defined by a Rigaku SmartLab X–Ray Diffractometer (XRD) (Rigaku Cooperation—Tokyo, Japan). The XRD measurements were carried out in the 2θ range between 40° and 54° at a scanning step of 0.01° using CuK$_\alpha$ radiation source. The compositional analysis was performed by an Energy Dispersive X–ray (EDX) spectroscopy. The X-ray diffraction beam is shined at the sample with a very narrow angle of incidence to increase the length of the X-ray beam that interacts with the thin film, keeping the sample stationary and rotating the receiver. Then the resulting diffraction beam appears on a concentric circle, recording the reflected beam intensity and first-order diffraction spectrum. An Oxford X–max 50 detector (Oxford Instruments, High Wycombe, UK) was used for the EDX measurements under an operating voltage of 20.00 kV. To study the surface structure, a Tescan MAIA3 Scanning Electron Microscopy (SEM) (TESCAN, Brno, Czech Republic) was used. The SEM measurements were done under the same operating voltage of 5.00 kV at room temperature. The particle sizes were determined from the SEM images using a freely available image processing and analysis software (ImageJ) (Software version for imageJ is 1.8.0). The roughness parameters were determined using a Veeco Multimode V Atomic Force Microscopy (AFM) (Veeco Instruments İnc., Santa Barbara, CA, USA) and evaluated using a WSxM 5.0 develop 9.4 software package [49]. To reveal the effect of the Co contents on the coercive field and squareness ratio, magnetic measurements were carried out by means of a JDAW–2000D model Vibrating Sample Magnetometer (VSM) (Xiamen Dexing Magnet Tech. Co., Ltd., City-Country: Xiamen, China) at ambient temperature and pressure by applying the external magnetic field parallel to the sample plane.

## 3. Results and Discussion

This paper aimed to study the impact of the Co contents on the structure, morphology, and magnetic characteristics of the Fe–Co–Ni deposits. To obtain the samples with various Co contents, the samples were grown onto ITO–coated glass substrates from PSs comprising different concentrations of Co ions using the electrochemical deposition technique. The potential–time transient curves are given in Figure 1.

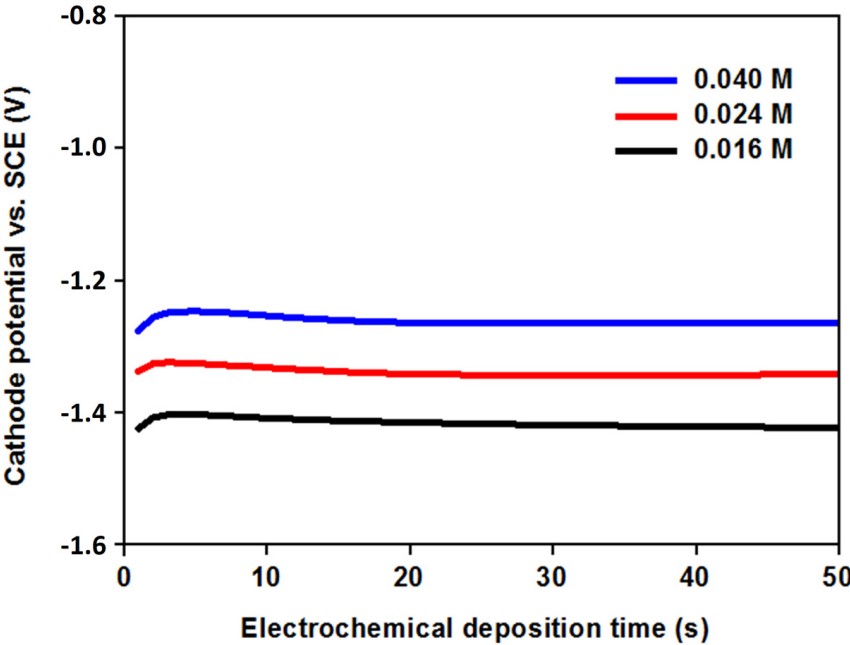

**Figure 1.** Potential–time transient curves of the samples electroplated at different ion concentrations of Co.

From Figure 1, it was understood that the samples can be grown properly from PSs containing different concentrations of Co ions owing to their stable cathode potentials. On the other hand, the cathode potential was detected to be higher for the sample electrochemically deposited from the PS with 0.016 M Co ion concentration compared to those determined for the samples deposited from PSs with 0.024 M and 0.040 M Co xml: confirmned with AE 1. keep email based on word 2. keep the ORCID 3. keep the format of not English for City in affs 4. keep the hyphen in figures 5. keep the two "Figure 6 indicates the AFM images of the samples. The samples possessed globular particles of various sizes" in textion concentrations.

EDX analyses showed that the samples electrochemically grown from PSs with different concentrations of Co ions have different Fe, Co, and Ni compositions. The compositional differences are shown in Table 1.

**Table 1.** The EDX data, phase structure, mean crystallite size, roughness parameters, coercive field, and squareness ratio of the samples.

|  | Co Ion Concentration (M) | | |
|---|---|---|---|
|  | **0.016** | **0.024** | **0.040** |
| Co (wt.%) | 32.9 | 43.2 | 60.9 |
| Ni (wt.%) | 62.4 | 52.7 | 35.7 |
| Fe (wt.%) | 4.7 | 4.1 | 3.4 |
| Resultant sample | $Fe–Co_{33}–Ni_{62}$ | $Fe–Co_{43}–Ni_{53}$ | $Fe–Co_{61}–Ni_{36}$ |
| Phase structure | fcc | fcc | Fcc + hcp |
| Mean crystallite size (nm) | 21.6 | 20.2 | 15.6 |
| RMS roughness (nm) | 14.4 | 17.8 | 28.4 |
| Average roughness (nm) | 11.0 | 14.0 | 21.8 |
| Average particle size (nm) | ~150 | ~14.0 | ~250 |
| Coercive field (Oe) | 36 | 51 | 121 |
| Squareness ratio (%) | 9.2 | 17.6 | 23.6 |

The sample grown from the PS with the lowest Co ion concentration of 0.016 M contained the lowest Co contents (32.9 wt.%), but the highest Ni (62.4 wt.%) and Fe (4.7 wt.%) compositions. In contrast to that, the sample electrochemically deposited from the PS with the highest Co ion concentration of 0.040 M included the highest Co contents (60.9 wt.%), but the lowest Ni (35.7 wt.%) and Fe (3.4 wt.%) compositions. The Co, Ni, and Fe compositions of the sample fabricated from the PS with an intermediate Co ion concentration of 0.024 M were 43.2, 52.7 and 4.1 wt.%, respectively. In summary, as the Co ion concentration in the PS was increased, the weight proportions of Ni and Fe components decreased, while the weight proportion of the Co component in the Fe–Co–Ni samples increased. Thus, three different ternary $Fe–Co_{33}–Ni_{62}$, $Fe–Co_{43}–Ni_{53}$ and $Fe–Co_{61}–Ni_{36}$ samples with different Co contents were fabricated. In recent studies [3,50], it was reported that the Co contents in electrochemically manufactured Ni–Co–Fe coatings and Co–Fe–Ni alloying micropillars increased but the Fe and Ni compositions decreased as the $Co^{2+}$ ion concentration in the PS increased. Similar results were also found in Fe–Co–Ni deposits electroplated in a former study [46], which was consistent with our results. On the other hand, in this work, the presence or absence of Anomalous Co–Deposition (ACD) was also explored. In this context, the relative Co, Ni, and Fe ion percentages in the PSs were compared to the relative Co, Ni, and Fe compositions in the samples.

As seen in Figure 2, the relative Co (Fe and Ni) composition in the sample increased with increasing relative Co (Fe and Ni) ion percentage in the PS. However, in all cases, the Co contents in the samples were determined to be higher than the Co ion percentage in the PSs (Figure 2a). The same phenomenon found for the Co component was also detected for

the Fe component (Figure 2b). This revealed the preferential electrochemical deposition for the Co and Fe components. However, the relative Ni composition in the samples was found to be lower than the relative Ni ion percentage in the PSs (Figure 2c). This indicated that the reduction of Ni components was inhibited. Thus, it was understood that the ACD behavior took place for all Co ions in the PS. The order of ACD was also revealed via composition ratio value (CRV).

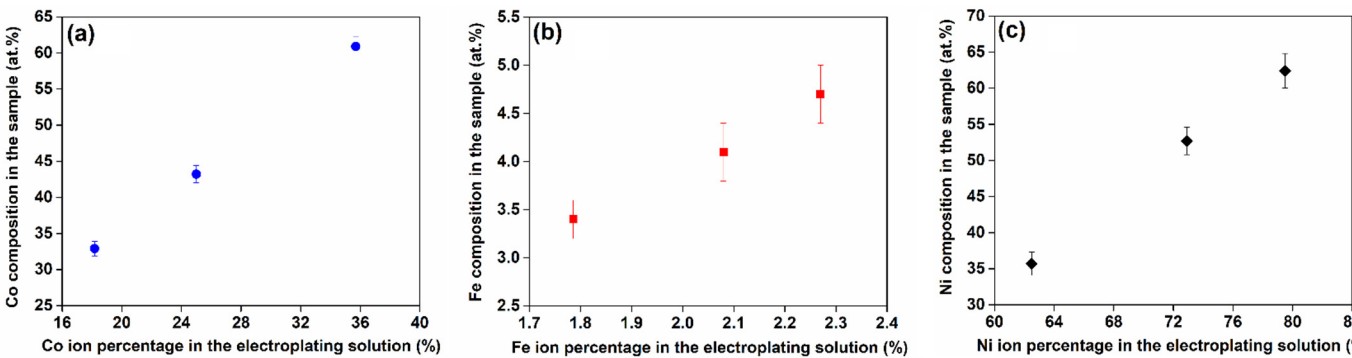

**Figure 2.** The relative Co (**a**), Fe (**b**), and Ni (**c**) composition in the samples against the relative Co (Fe and Ni) ion percentage in the PSs.

The CRV for a Co component is described by the following expression [46].

$$\text{CRV for Co} = \frac{\text{relative composition of Co in the sample}}{\text{relative ion concentration of Co in the PS}} \quad (1)$$

The relative concentration of Co ions in the PS is given by the following expression:

$$\text{Relative concentration of Co ions} = \frac{[CoSO_4]}{[CoSO_4 + NiSO_4 + FeSO_4]} \times 100 \quad (2)$$

The above procedure was also applied to calculate the $CRV_{Fe}$ and $CRV_{Ni}$. From the results of the analysis depicted in Figure 3, it was understood that the $CRV_{Fe}$ and $CRV_{Co}$ were higher than one, while the $CRV_{Ni}$ was lower than one, revealing that the reduction rate of $Ni^{2+}$ was significantly lower than the reduction rates of $Fe^{2+}$ and $Co^{2+}$ during the deposition process. This phenomenon confirmed the creation of ACD behavior, which is the characteristic feature for the electrochemical deposition of iron–group alloys. Furthermore, the reduction rate of $Fe^{2+}$ was higher compared to the reduction rate of $Co^{2+}$ as the $CRV_{Co}$ was lower than $CRV_{Fe}$ [3,6,19,24,46,51–53]. Therefore, the degree of ACD characteristics of Co–Ni was lower than Fe–Ni. In addition, the $CRV_{Fe}/CRV_{Co}$ ratio was determined to be lower than the $CRV_{Co}/CRV_{Ni}$ ratio, revealing that the order of ACD was Fe–Ni > Co–Ni > Fe–Co for all Co ion concentrations in the PS, which is in good agreement with the findings of conducted studies [3,6,24,46].

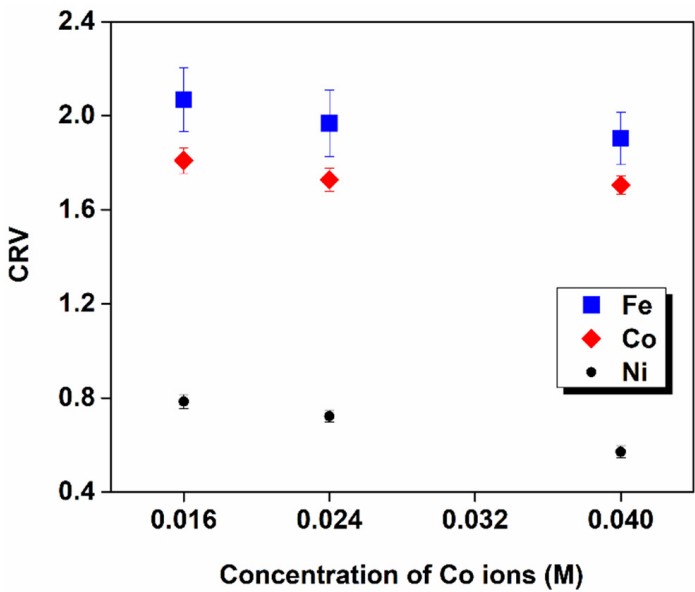

**Figure 3.** The CRV$_{Fe}$, CRV$_{Co}$ and CRV$_{Ni}$ as a function of the Co ion concentration in the PS.

The phase structure of the samples produced in this work was investigated via XRD analysis. The resulting XRD patterns are shown in Figure 4.

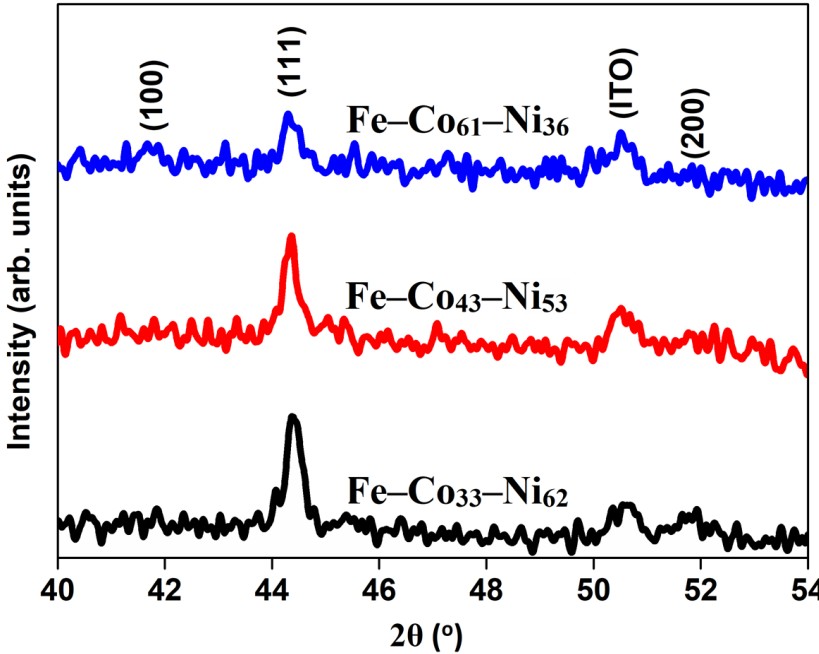

**Figure 4.** XRD patterns of the samples.

Obviously, in all samples, the (111) diffraction peak of the face–centered cubic (fcc) crystal structure observed at the angular position of about 2θ = 44.4° was the most intense irrespective of the Co contents. The single phase structure (fcc) obtained for the Fe–Co$_{33}$–Ni$_{62}$ and Fe–Co$_{43}$–Ni$_{53}$ samples shows a good agreement with the XRD patterns of the ternary ferromagnetic materials with similar compositions produced in previous studies [6,21,24,54,55]. In addition to that, compared to the Fe–Co$_{33}$–Ni$_{62}$ and Fe–Co$_{43}$–Ni$_{53}$ samples, the XRD pattern of the Fe–Co$_{61}$–Ni$_{36}$ sample also revealed the presence of the (100) diffraction peak with low intensity related to the hexagonal close–packed (hcp) phase structure which occurred at about 2θ = 41.7°. At high Co contents, a transition from single phase structure (fcc) to dual phase structure (fcc + hcp or fcc + bcc) was also reported

in electrochemically grown binary Ni–Co films, Co–Ni–Al$_2$O$_3$ composite coatings, and ternary ferromagnetic films of Fe, Co, and Ni [12,13,17,47,56]. Alongside the phase transition, an increment in the Co contents resulted in a significant decrement in the intensities of both (111) and (200) diffraction peaks, reflecting a strong reduction in the crystallization (Figure 4). This also caused a change in the crystallite size of the samples. The crystallite size (D) of the produced samples was determined by Scherrer's equation [57]:

$$D = [0.9\lambda/Bcos\theta] \times [180°/\pi] \tag{3}$$

where λ, B and θ represent the wavelength of CuKα radiation, Full–Width at Half Maximum (FWHM) value, and Bragg diffraction angle, respectively. To estimate the B and θ values, XRD patterns were fitted by Lorentzian curves. It was revealed that mean crystallite size decreases from 21.6 to 15.6 nm as the Co contents in the samples increases from 33 to 61 wt.%, indicating that the crystallite size of the Co–rich samples is smaller compared to the Co–poor samples. The decrease in the crystallite size with the Co contents was also reported in Fe–Co–Ni films electroplated on copper substrates from ammonium–chloride–based PSs [47]. The cause of the increase (decrease) in the size of Fe–Co–Ni is due to the lattice constant of Fe, Co, and Ni atoms and the interaction between electrons leads to the appearance of size effect. The 3-D surface microtexture can be characterized for a deeper understanding of the nano-scale patterns by stereometric [58–60] and fractal/multifractal analyses [61–64].

The surface topography was studied by means of SEM images analysis of the samples. The SEM device we used has a resolution of 100 KX, the crystals can be observed in Figure 5.

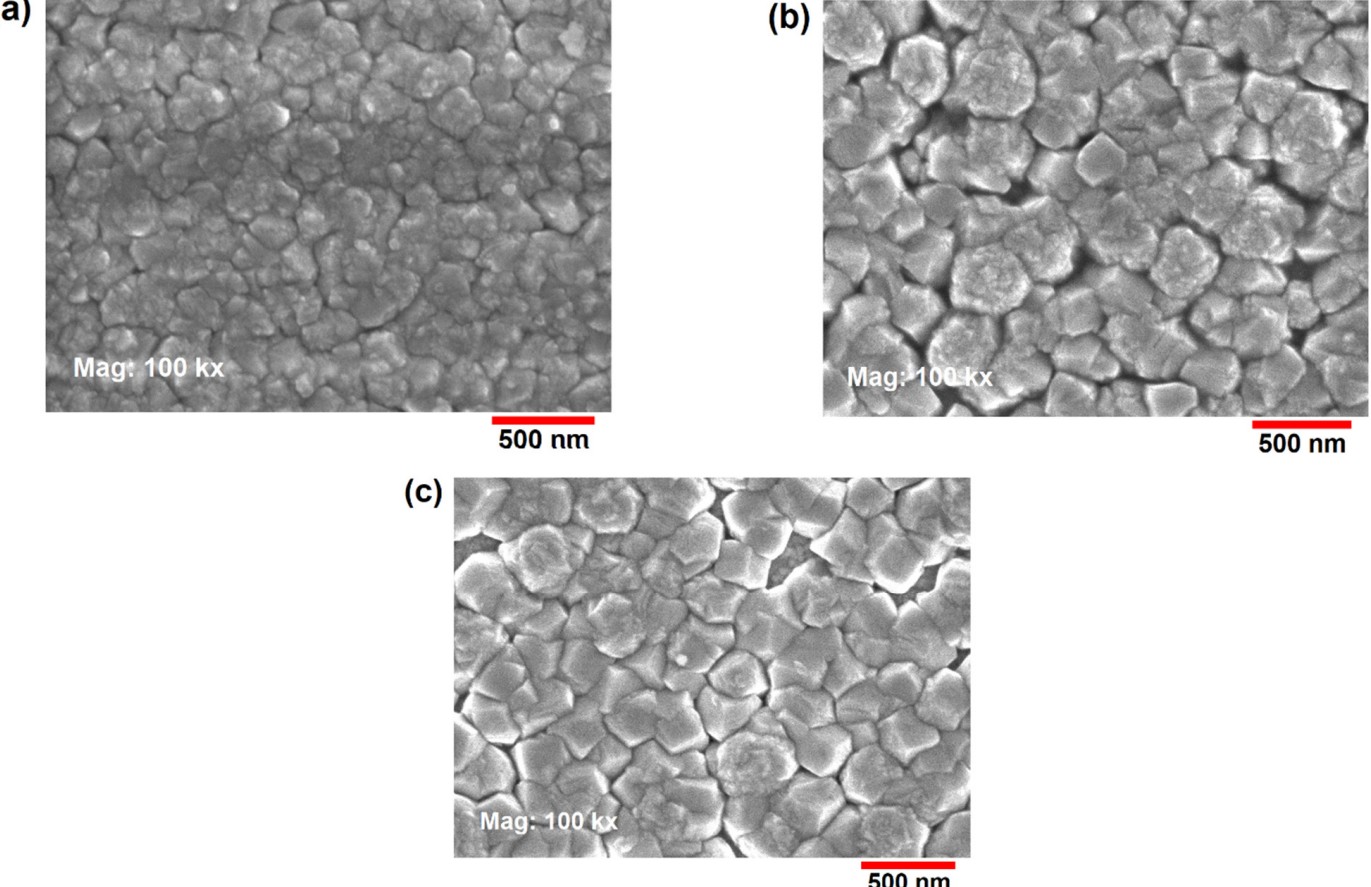

**Figure 5.** SEM images of the samples Fe–Co$_{33}$–Ni$_{62}$ (**a**), Fe–Co$_{43}$–Ni$_{53}$ (**b**), and Fe–Co$_{61}$–Ni$_{36}$ (**c**), respectively.

Figure 5a showed that the Fe–$Co_{33}$–$Ni_{62}$ sample exhibits a surface topography comprising only cauliflower–like agglomerates.

However, as seen from Figure 5b,c, the Fe–$Co_{43}$–$Ni_{53}$ and Fe–$Co_{61}$–$Ni_{36}$ samples had a surface topography covered with a mixture of pyramidal particles and cauliflower–like agglomerates. The surface morphology of a material is due to the lattice constant, the electronic interactions between different atoms caused.

In all samples, one cauliflower–like agglomerate was composed of grains. An increment in the Co contents caused an enhancement in the size of the cauliflower–like agglomerates and a decrease in their number. The average width of the cauliflower–like agglomerates was found to be about 150 nm for the Fe–$Co_{33}$–$Ni_{62}$ sample. However, the Fe–$Co_{43}$–$Ni_{53}$ and Fe–$Co_{61}$–$Ni_{36}$ samples had larger cauliflower–like agglomerates with an average width of about 270 and 350 nm, respectively. The size of the crystal nuclei is larger than the average size of the particles. The cause of this phenomenon is due to the difference between the lattice constants and the interactions between the electronic structures which leads to the formation of nuclei crystallization in the form of cauliflower.

On the other hand, the average width of the pyramidal particles was determined to be about 190 and 220 nm for the Fe–$Co_{43}$–$Ni_{53}$ and Fe–$Co_{61}$–$Ni_{36}$ samples, respectively. Thus, the Fe–$Co_{61}$–$Ni_{36}$ sample exhibited larger pyramidal particles than the Fe–$Co_{43}$–$Ni_{53}$ sample. Consequently, as also listed in Table 1, the Fe–$Co_{33}$–$Ni_{62}$, Fe–$Co_{43}$–$Ni_{53}$ and Fe–$Co_{61}$–$Ni_{36}$ samples had particles with an average width of approximately 150, 210, and 250 nm, respectively, which was in good agreement with the findings reported in the literature [47]. Further studies on the morphological characteristics were also carried out using an AFM. Figure 6 indicates the AFM images of the samples. The samples possessed globular particles of various sizes. The size of globular particles increased and their number decreased when the Co content increased, which is consistent with the findings of the SEM analysis. The surface morphology of the material is caused by the lattice constant, the electronic force of interaction between the atoms of the material. On the other hand, the influence of the Co content on the particle size can be ascribed to different cathode potentials caused by the concentration of Co ions in the PS. Conducted studies showed that the particle size decreased when the cathode potential was increased [24].

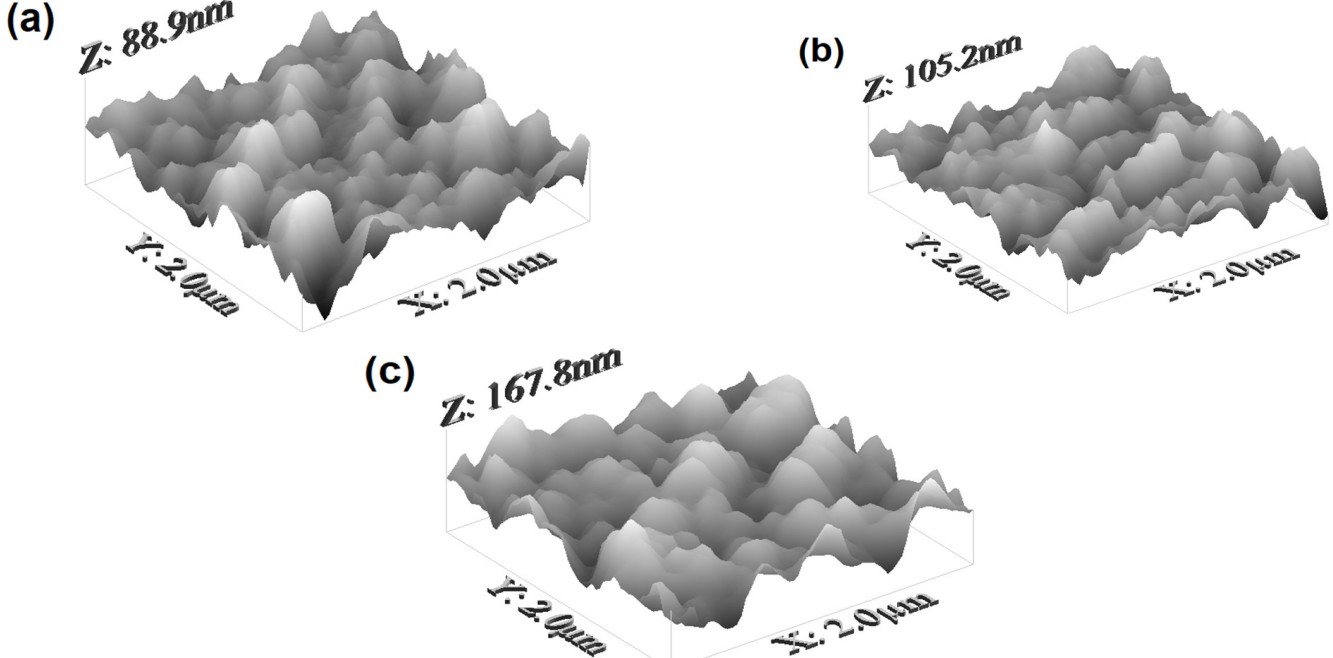

**Figure 6.** AFM images of the samples (**a**) Fe–$Co_{33}$–$Ni_{62}$, (**b**) Fe–$Co_{43}$–$Ni_{53}$, and (**c**) Fe–$Co_{61}$–$Ni_{36}$, respectively.

Therefore, the particle size increased with the Co content since the cathode potential decreased with increasing Co ion concentration in the PS (Figure 1). The roughness parameters were determined from the AFM images. As distinctly noticed in Table 1, the Co content had a significant effect on the surface roughness parameters. When the Co content in the samples increases, the electronic interaction of Co with Fe and Ni increases, so the size of crystal nuclei increases, and the number of spheres is reduced leading to the increased surface roughness of the material.

Figure 6 indicates the AFM images of the samples. The samples possessed globular particles of various sizes.

The obtained results revealed that the surface roughness increased the cause is due to when Co content increased, leading to the electronic interaction of Co atoms with Fe, Ni increases, and the size of crystal nuclei increases and the number of spheres reduces to increased surface roughness of the material.

In this paper, the thickness and size of the considered thin films are not investigated especially, but the thickness of the thin film has an average size equal to the size of the crystal nuclei such as: 150 nm with $Fe–Co_{33}–Ni_{62}$, 270 nm with $Fe–Co_{43}–Ni_{53}$, and 350 nm with $Fe–Co_{61}–Ni_{36}$.

Figure 7 exhibits the normalized in–plane magnetic hysteresis loops measured to determine the magnetic characteristics with respect to their Co contents. The results obtained from the magnetic analysis are collected in Table 1. Although all samples exhibited a ferromagnetic behavior with a magnetic hardness being between soft and hard [3,17,21,22,24,55,65], the Co contents played a significant role in the coercive field. The coercive field increased considerably from 36 to 121 Oe as the Co contents in the samples increased from 33 to 61 wt.%. Increasing the content of Co leads to increased crystallization, increasing the size of the crystal nuclei (magnetic domain) leading to an increase in the coercive field of the material, which is shown in Figure 7. The increase in the coercive field with the Co content may also be attributed to an increment in the surface roughness.

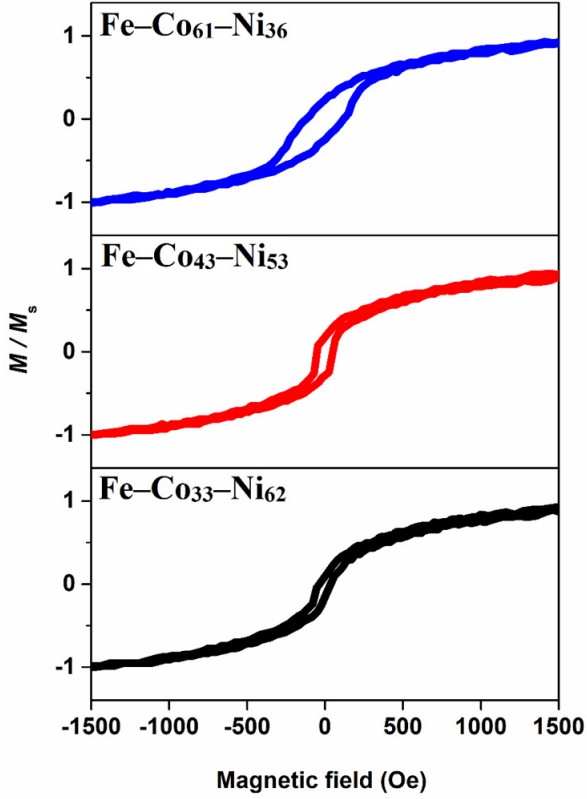

**Figure 7.** Hysteresis loops of the samples with different compositions.

The variations observed in the average surface roughness and coercive field with respect to the Co contents are shown in Figure 8.

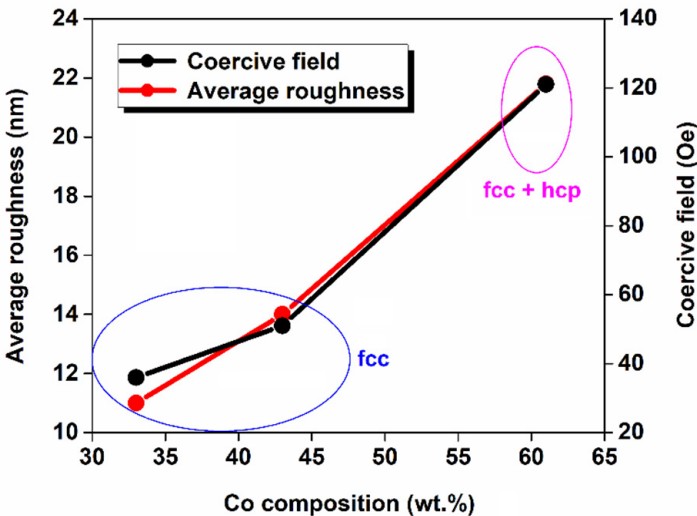

**Figure 8.** The correlation between the coercive field and the surface roughness.

As clearly evidenced in Figure 8, there was a direct correlation between the surface roughness and the coercive field as also reported in the electrochemically deposited ternary ferromagnetic films of Fe, Co, and Ni [21,24,65,66]. In addiiton, the Fe–$C_{o61}$–$Ni_{36}$ sample had a much higher coercive field than the Fe–$Co_{33}$–$Ni_{62}$ and Fe–$Co_{43}$–$Ni_{53}$ samples. This abrupt increase in the coercive field with increasing Co contents from 43 to 61 wt.% may also be ascribed to the appearance of the (100) diffraction peak of the hcp phase structure of Co. In previous studies [13,17,47], it was shown that the Ni–Co and Fe–Co–Ni films with single phase structure at low Co contents exhibit a much lower coercive field than the Ni–Co and Fe–Co–Ni films with dual phase structure at high Co contents. On the other hand, the produced samples were found to have very low squareness ratios ranging from 9.2 to 23.6%. Such low squareness ratios correspond to the formation of an in–plane hysteresis loop with a vertical magnetization component as also observed in electroplated Ni–Co/ITO and Fe–Co–Ni/ITO thin film samples [21,65]. Furthermore, as clearly seen from Table 1, a gradual increment in the magnetic squareness ratio was detected with the Co contents, revealing a decrement in the vertical component of magnetization.

## 4. Conclusions

This work aimed to obtain the ternary Fe–Co–Ni samples with various Co contents and reveal the differences in the morphological, magnetic, and structural properties with respect to their Co contents. According to the compositional analysis, a change in the Co ion concentration of the PS significantly affected the deposit composition. It was understood that the Fe–Co–Ni sample with higher Co contents could be obtained when the sample was electrochemically deposited from the PS including higher Co ion concentration. It was also revealed that the co–deposition characteristic (anomalous) and its order (Fe–Ni > Co–Ni > Fe–Co) were not affected by the amount of Co ions in the PS. The resultant samples exhibited the predominant reflection from the (111) plane of the fcc crystal structure. Unlike the Fe–$Co_{33}$–$Ni_{62}$ and Fe–$Co_{43}$–$Ni_{53}$ samples, the Fe–$Co_{61}$–$Ni_{36}$ sample with the highest Co contents exhibited a weak reflection from the (100) plane of the hcp crystal structure of Co. Compared to the Fe–$Co_{61}$–$Ni_{36}$ and Fe–$Co_{43}$–$Ni_{53}$ samples, the crystallinity was found to be stronger for the Fe–$Co_{33}$–$Ni_{62}$ sample. The size of the crystallites decreased from 21.6 to 15.6 nm as the Co contents in the sample increased from 33 to 61 wt.%. A surface structure covered with a mixture of pyramidal particles and cauliflower–like agglomerates was detected for the Fe–$Co_{43}$–$Ni_{53}$ and Fe–$Co_{61}$–$Ni_{36}$ samples, whereas the Fe–$Co_{33}$–$Ni_{62}$ sample had a surface structure consisting of only cauliflower–like agglomerates.

Moreover, compared to others, the Fe–Co$_{33}$–Ni$_{62}$ sample exhibited a more compact surface morphology consisting of smaller cauliflower–like agglomerates with an average width of 150 nm. As the Co contents enhanced, the average and RMS surface roughness parameters increased significantly from 11.0 to 21.8 nm and from 14.4 to 28.4 nm, respectively. The samples produced were magnetically semi–hard. However, the Fe–Co$_{61}$–Ni$_{36}$ (121 Oe) thin film sample exhibited a noticeably higher coercive field compared to the Fe–Co$_{43}$–Ni$_{53}$ (51 Oe) and Fe–Co$_{33}$–Ni$_{62}$ (36 Oe) thin film samples, which was attributed to the phase transition from single phase structure (fcc) to dual phase structure (fcc + hcp) and an abrupt enhancement in the surface roughness parameters. An increase in the Co contents from 33 to 61 wt.% also induced an enhancement in the magnetic squareness ratio from 9.2 to 23.6%, reflecting a decrement in the vertical component of magnetization.

**Author Contributions:** V.C.L.: Validation, writing and editing. U.S.: conceptualization, methodology, investigation. M.C.B.: validation, writing and editing. L.D.T.: writing and editing. Ş.Ṭ.: writing–editing. D.N.T.: conceptualization, methodology, investigation, resources, supervision, writing—original draft—review and editing. All authors have read and agreed to the published version of the manuscript.

**Funding:** This research was financially supported by the Scientific Research Projects Commission of Bartın University under the project number 2018–FEN–A–021.

**Institutional Review Board Statement:** Not applicable.

**Informed Consent Statement:** Not applicable.

**Data Availability Statement:** The data that support the findings of this study are available from the corresponding author upon reasonable request.

**Acknowledgments:** The authors would like to thank Çağdaş Denizli for taking AFM images and Malik Kaya for his assistance in the electrodeposition process of the samples.

**Conflicts of Interest:** The authors declare no conflict of interest. The funders had no role in the design of the study; in the collection, analyses, or interpretation of data; in the writing of the manuscript, or in the decision to publish the results. Neither author has a financial or proprietary interest in any material or method mentioned.

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
