# Peer review of "Electrochemical Deposition of Fe–Co–Ni Samples with Different Co Contents and Characterization of Their Microstructural and Magnetic Properties"

_coatings, doi:10.3390/coatings12030346_

Round 1

Reviewer 1 Report

  1. Please indicate the highlights of the manuscript.
  2. There are many words or expressions that need to be improved in this manuscript, e.g., in the abstract or the title, the Co composition should be Co content, etc.
  3. Why is the average particle size larger than the average grain size?
  4. What factors contribute to the formation of a surface topography with different particle sizes?
  5. Why does the surface roughness derived from the measurements carried out using an AFM significantly increase as the Co content in the samples increased?
  6. why does the coercive field increase considerably from 36 to 121 Oe as the Co content in the samples increased from 33 to 61 wt.%?
  7. Please indicate the difference in the properties for the thin film fabricated by at least two different methods to indicate the merits of this work.

Author Response

Review 1

Answer

The author thanks the reviewer for taking the valuable time to read the content of the manuscript, and for the helpful comments that gave the author the opportunity to edit and improve the content of the manuscript. The following responses are intended to answer all questions of the reviewer and to finalize the content of the manuscript.

Please indicate the highlights of the manuscript.

Answer

-Fe-Co-Ni materials have strong reflectors at (111) and weak at (100) of the fcc crystal structure.

- The crystallinity size of Fe–Co33–Ni62 is large and decreases from 21.6 to 15.6 nm when Co increases from 33 to 61% by weight.

-When Co is increased, the average surface roughness and RMS parameters increase from 11.0 to 21.8 nm and from 14.4 to 28.4 nm, respectively.

-When Co increased from 33 to 61%, the squared ratio of magnetism increased from 9.2 to 23.6%, reflecting the decrease of magnetization.

  1. There are many words or expressions that need to be improved in this manuscript, e.g., in the abstract or the title, the Co composition should be Co content, etc.

Answer

We replaced the word "Co composition" by “Co content”.

  1. Why is the average particle size larger than the average grain size?

Answer

We edited the sentence "the size of the crystal nuclei is larger than the average size of the particles". The cause of the problem phenomenon is due to the difference between the lattice constants and the interactions between the electronic structures; in consequence we have the formation of cauliflower crystals.

3.What factors contribute to the formation of a surface topography with different particle sizes?

Answer

The surface morphology of the material is caused by the lattice constant, the electronic force of interaction between the atoms of the material. On the other hand, the influence of the Co content on the particle size can be ascribed to different current densities caused by the concentration of Co ions in the PS. Conducted studies showed that the particle size decreased when the cathode potential (or applied current density) was increased [24]. Therefore, the particle size increased with the Co content since the cathode potential decreased with increasing Co ion concentration in the PS (Fig. 1).

  1. Why does the surface roughness derived from the measurements carried out using an AFM significantly increase as the Co content in the samples increased?

Answer

AFM is a specialized tool for determining the micro/nano-patterns of the surface roughness of materials. The roughness parameters were determined from the Atomic Force Microscope (AFM) images. As distinctly noticed in Table 1, the Co content had a significant effect on the surface roughness parameters. The obtained results revealed that the surface roughness derived from the measurements carried out using an AFM significantly increased as the Co content in the samples increased.

When Co content in the samples increased, the electronic interaction of Co with Fe, Ni increases, so the size of crystal nuclei increases, and the number of spheres is reduced leading to increased surface roughness of the material.

  1. Why does the coercive field increase considerably from 36 to 121 Oe as the Co content in the samples increased from 33 to 61 wt.%?

Answer

Because increasing the Co content leads to increased crystallization, increasing the size of the crystal nuclei (magnetic domain) leads to an increase in the residual coercivity and the coercivity of the material, what is shown in Fig. 6.

  1. Please indicate the difference in the properties for the thin film fabricated by at least two different methods to indicate the merits of this work.

Answer

To make magnetic Fe-Co-Ni thin films, researchers can use many methods such as: Evaporation, electrochemical deposition [48] etc. By using electrochemical deposition method, the thin film is obtained with a high uniformity, according the following reference:

[48] ​​F. Omata, Magnetic properties of Fe-Co-Ni films with high saturation magnetization prepared by evaporation and electrodeposition, Ieee translation Journal on magnetics in Japan, 1990, 5, 17-28.

Reviewer 2 Report

Introduction is well written and documented. Experiments are clearly presented and analysis is sound. The work will certainly deserve publication after consideration of following points.

  1. I cannot find any indication of the overall thickness of the thin films. Authors should estimate this average thickness, which is indeed useful information for the appreciation of AFM z-profiles in figure 6 and SEM micrographs in fig. 5.
  2. In lines 224-225, authors state that "one cauliflower–like agglomerate was composed of many smaller grains". What is the range of this smaller grain size? How does the grain size compare with the mean-crystallite size from Scherrer equation?

2a. What is the resolution of the SEM experiments? Could the crystallites be observable?

  1. What is scan mode of XRD experiments in figure 4? Please, complete information on scan conditions in line 101.
  2. To calculate crystallite size, authors fitted XRD patterns with Gaussian curves. It is their right. However, in their precedent work in ref 21, they used Lorentzian curves for a very similar system. So, authors should give a comment about their choice of fit type.
  3. Authors should consider adding a plot of fit profiles like the figure 2c of their precedent work in ref 21. It helps readers to directly compare reflection positions and widths.

6. Authors adequately cite many references on similar coatings including their previous works. Moreover, they claim that "The obtained results have very high applicability in life and are the basis for the research and fabrication of future magnetic devices." So, how do their new results compare with previous systems in literature? Authors need to add some comments for comparison.

Author Response

Review 2

The author thanks the reviewer for taking valuable time to read the content of the manuscript, the comments are very helpful to help the author have a better opportunity to edit the manuscript content. The following responses are intended to answer all questions of the reviewer.

  1. I cannot find any indication of the overall thickness of the thin films. Authors should estimate this average thickness, which is indeed useful information for the appreciation of AFM z-profiles in figure 6 and SEM micrographs in fig. 5.

Answer

According with the content of the article, the authors did not investigate especially the thickness and size of considered thin films, but in our paper the thickness of the thin film has an average size equal to the size of the crystal nuclei such as: 150 nm with Fe-Co33-Ni62, 270 nm with Fe-Co43-Ni53 and 350 nm with Fe-Co61-Ni36.

  1. In lines 224-225, authors state that "one cauliflower–like agglomerate was composed of many smaller grains". What is the range of this smaller grain size? How does the grain size compare with the mean-crystallite size from Scherrer equation?

Answer

We edited the sentence "one cauliflower–like agglomerate was composed of many smaller grains" to "one cauliflower–like agglomerate was composed of many grains" to avoid misunderstanding.

The average sizes of the crystal nuclei are 150 nm with Fe-Co33-Ni62, 270 with Fe-Co43-Ni53 and 350 nm with Fe-Co61-Ni36.

While, the average crystal size of the crystal nuclei calculated from Scherrer's equation showed size from 21.6 to 15.6 nm when Co content increases from 33 to 61 wt.%

  1. What is the resolution of the SEM experiments? Could the crystallites be observable?

Answer

The resolution of the SEM device has a resolution of 100 KX, and the crystals can be observed as shown in Figure 5.

  1. What is scan mode of XRD experiments in figure 4? Please, complete information on scan conditions in line 101.

Answer

The XRD measurements were carried out in the 2θ range of between 40–54 degrees at a scanning step of 0.01 degrees using CuKα radiation source. We shine the X-ray diffraction beam at the sample with a very narrow angle of incidence to increase the length of the X-ray beam which interacts with the thin film, keeping the sample stationary and rotating the receiver. Then the resulting diffraction beam appears on a concentric circle, recording the reflected beam intensity and first order diffraction spectrum.

  1. To calculate crystallite size, authors fitted XRD patterns with Gaussian curves. It is their right. However, in their precedent work in ref 21, they used Lorentzian curves for a very similar system. So, authors should give a comment about their choice of fit type.

Answer

We edited the sentence “XRD patterns with Gaussian curves” to “XRD patterns with Lorentzian curves”.

  1. Authors should consider adding a plot of fit profiles like the figure 2c of their precedent work in ref 21. It helps readers to directly compare reflection positions and widths.

Answer

The author thanks the reviewer for pointing out what is lacking compared to our precedent paper [21], but the results given in Figure 4 in the content of this manuscript also fully demonstrate those characteristics mentioned by the Reviewer, in our opinion it is not necessary to present details described in paper [21], we hope that this receives the approval of the reviewer.

  1. Authors adequately cite many references on similar coatings including their previous works. Moreover, they claim that "The obtained results have very high applicability in life and are the basis for the research and fabrication of future magnetic devices." So, how do their new results compare with previous systems in literature? Authors need to add some comments for comparison.

Answer

The author thanks the reviewer for pointing out this unfortunate mistake caused by incorrect word usage leading to misunderstanding. Author would like to revise that content from “The obtained results have very high applicability in life and are the basis for the research and fabrication of future magnetic devices.” to “The obtained results would be applied in different fields of science and technology.”

Reviewer 3 Report

In this manuscript, the authors successfully electroplated Fe-Co-Ni samples on indium thin oxide substrates, where the Co composition could be tuned by adjusting the amount of Co ion concentration in the plating solutions. Their structure, morphology, and magnetic characteristics as the function of Co composition were systematically investigated. The results are very comprehensive and convincing. I will suggest its publication in Coatings in present form.

Author Response

Review 3

In this manuscript, the authors successfully electroplated Fe-Co-Ni samples on indium thin oxide substrates, where the Co composition could be tuned by adjusting the amount of Co ion concentration in the plating solutions. Their structure, morphology, and magnetic characteristics as the function of Co composition were systematically investigated. The results are very comprehensive and convincing. I will suggest its publication in Coatings in present form.

Answer

Thanks to the reviewer for taking valuable time to read the manuscript content and give us very good opinion about our paper.

Round 2

Reviewer 1 Report

(1) In Figure 2, remove the brackets and leave Fe and Ni. (2) In line 224 , The sentence "The SEM images of the samples."should be deleted. (3) In lines 280-281, please indicate the references to show the basis of the author's argument.

Reviewer 2 Report

Authors added the missing information and adequately revised their manuscript. I regret that results are still not compared with previous systems in literature (point 6) but this is acceptable with the revision made. Therefore, I recommend acceptation of the manuscript in present form.